# Evaluation of Grading Estrogen Receptors in Breast Cancer Using Fully Automated Rapid Immunohistochemistry Based on Alternating-Current Electric Field Technology

**DOI:** 10.3390/cancers17030363

**Published:** 2025-01-23

**Authors:** Chiaki Kudo, Kaori Terata, Hiroshi Nanjo, Kyoko Nomura, Yuko Hiroshima, Eriko Takahashi, Ayuko Yamaguchi, Hikari Konno, Masaaki Onji, Yuki Wakamatsu, Yoshihiko Kimura, Shinogu Takashima, Akiyuki Wakita, Yusuke Sato, Yoshihiro Minamiya, Kazuhiro Imai

**Affiliations:** 1Department of Thoracic Surgery, Akita University Graduate School of Medicine, Akita 010-8543, Japan; c19my92k@yahoo.co.jp (C.K.); keriri918@gmail.com (E.T.); hkakh121@gmail.com (A.Y.); konno1227h@yahoo.co.jp (H.K.); masaaki53onji68@gmail.com (M.O.); ywakamatsu@gipc.akita-u.ac.jp (Y.W.); s.takashima0919@gmail.com (S.T.); wakita@gipc.akita-u.ac.jp (A.W.); yusuke@doc.med.akita-u.ac.jp (Y.S.); minamiya@med.akita-u.ac.jp (Y.M.); karo@doc.med.akita-u.ac.jp (K.I.); 2Department of Breast and Endocrine Surgery, Akita University Hospital, Akita 010-8543, Japan; 3Department of Thoracic and Breast Surgery, Akita Kousei Medical Center, Akita 011-0948, Japan; ykimura@js7.so-net.ne.jp; 4Department of Pathology, Akita University Hospital, Akita 010-8543, Japan; hnanjo@med.akita-u.ac.jp (H.N.); yuko-kitty@hos.akita-u.ac.jp (Y.H.); 5Department of Environmental Health Science and Public Health, Akita University Graduate School of Medicine, Akita 010-8543, Japan; nomurakyoko@gmail.com

**Keywords:** immunohistochemistry, rapid immunohistochemistry, breast cancer, estrogen receptor

## Abstract

We previously developed a rapid immunohistochemistry (R-IHC) method based on alternating-current electric field technology and have now developed a fully automated rapid IHC stainer (R-Auto). We evaluated 188 breast cancer surgical specimens at our hospital via estrogen receptor (ER) staining using R-Auto, conventional IHC, and a commercial autostainer. The specimens were scored using Allred scores, after which the staining results were compared between R-Auto and conventional IHC or the commercial autostainer. The AC_1_ statistic for comparison between R-Auto and conventional IHC and R-Auto was 0.9490, with a 95.7% agreement rate, and that for comparison between R-Auto and the commercial autostainer was 0.9095, with a 92.6% agreement rate. There was substantial agreement between R-Auto and each procedure. Furthermore, R-Auto shortened the time required for IHC from 210 min with the commercial autostainer and 209 min with conventional IHC to 42 min.

## 1. Introduction

The immunohistochemistry (IHC) technique uses antibodies to detect specific antigens within samples. Combining antigen–antibody reactions with a color reaction enables the presence and localization of an antigen within a tissue sample to be observed under a microscope, thereby confirming the expression of specific genes or proteins. For cancer treatment, IHC helps make differential diagnoses of benign and malignant tumors, visualize the extent of cancers, and select drug therapies. However, standard IHC procedures in frozen or paraffin sections take 2–4 h, so they cannot be used for intraoperative frozen section diagnosis [1,2,3].

We developed an innovative device that enables the rapid completion of IHC analyses within approximately 20 min. With this device, we apply a high-voltage, low-frequency alternating-current (AC) electric field to tissue sections. The resultant coulomb force stirs the antibody solution within droplets on the sections without increasing the temperature of the droplets. We previously reported the utility of this method for staining frozen sections for the intraoperative detection of lymph node metastasis in non-small cell lung cancer [4]; the differential diagnosis of brain [5] and lung tumors [6]; and the detection of sentinel lymph node metastasis in breast cancer [7]. The activated antigen–antibody reaction induced with this method could also reduce the amount of the expensive antibody needed for analysis [8]. Similarly, there have been reports on developments related to rapid IHC. For example, the literature reported that the rapid IHC method using the intermittent microwave irradiation method (MW) makes it possible to reduce the entire immunohistochemical staining process to one hour [9]. However, because this method uses microwaves, there is concern about tissue and cell degeneration due to high-temperature heat. Our technology has an advantage in this respect.

As originally developed, our rapid IHC (R-IHC) method was largely manual. Only the antibody reaction was carried out in a staining machine, and the application of reagents and washing out of antibodies were carried out manually. We developed a fully automated R-IHC staining device (R-Auto) that automates all steps in the IHC protocol to streamline the protocol and make it more reproducible. The entire staining process can be carried out automatically by filling the cartridge with the necessary reagents, registering them in the instrument, and setting the protocol. Moreover, the AC electric field is applied to the process of washing out the antibody solutions, which was not performed when washouts were carried out manually.

We previously reported that R-IHC is a time-saving utility useful for intraoperative frozen sections. Meanwhile, formalin-fixed paraffin-embedded (FFPE) sections are crucial for diagnosis and treatment decision making in nearly all cancer patients, regardless of the disease stage, and many diagnoses are made using FFPE sections in daily practice. Against that background, in this report, we evaluated the utility of R-Auto for use with FFPE specimens of breast cancer, the most commonly occurring cancer in women worldwide [10]. IHC analyses of the estrogen receptor (ER), the progesterone receptor (PgR), human epidermal growth factor receptor type 2 (HER2), and the Ki-67 index were used as surrogates for genetic profiling to determine breast cancer treatment strategies, which include neoadjuvant or adjuvant systemic therapy to reduce the risk of recurrence, to prolong life and maintain quality of life. IHC has contributed to the remarkable progress in breast cancer therapeutics by enabling the assembly of the most appropriate treatments against specific forms of the disease.

This study aimed to establish a staining protocol and evaluate the staining performance of R-Auto against ERs as a prototype for breast cancer FFPE specimens.

## 2. Methods

### 2.1. Ethical Conditions and Patient Samples

The medical records of 200 patients who underwent surgery for breast cancer at our hospital between July 2014 and January 2017 were serially extracted. Among them, 188 eligible patients with appropriate pathological specimens were enrolled in this study. This retrospective study was approved by the Institutional Review Board at the Akita University School of Medicine and University Hospital (permit number: 3105). All samples were collected under IRB Protocol No. 3105. A diagram of the case selection process for this study is shown in Figure 1.

### 2.2. IHC Procedures

The breast cancer specimens were fixed in 10% buffered formalin and embedded in paraffin. Four-micrometer-thick sections from the FFPE samples were incubated with Paraffin Stretcher (Sakura Finetek Japan Co., Ltd., Tokyo, Japan) at 50 °C overnight and then stained using IHC. At the time of their surgery, each patient’s FFPE specimen was stained for ER using IHC performed with a commercially available autostainer (Ventana BenchMark ULTRA (Ventana Medical Systems, Tucson, AZ, USA)). In this study, staining for ER was also carried out using R-Auto and conventional manual (hereafter referred to as conventional) IHC.

### 2.3. R-IHC Method and R-Auto Protocol

We developed a device that reduces the time required for IHC analysis. Its mechanism has been previously described in detail [6,11]. Briefly, in this R-IHC system, a high-voltage, low-frequency (4 KV; 10 Hz) alternating-current (AC) electric field is applied to tissue sections while they are incubating with the antibodies. A schematic diagram of the device used to apply the high-voltage, low-frequency AC electric field is shown (Figure 2a,b). The slide is placed between the electrodes, and a high-voltage (4 KV), low-frequency (5 Hz) AC current is applied. The arrows in each figure indicate that the antibodies are being stirred due to the vibration of the droplets. “Schema of the device used to apply a high-voltage, low-frequency AC electric field. The slide was placed between the electrodes, and a high-voltage (4 KV), low-frequency (5 Hz) AC current was applied. The schema shows the changes within a microdroplet as the voltage is switched on and off in a time series (I → II → III → IV), which mixes the antibodies” (this sentence is cited from reference [7]) (Figure 2b). This significantly shortens the time required for the antigen–antibody reaction. The AC electric field is effective for stirring liquids in the order of µL and is an ideal technique for IHC as it does not increase the temperature of the droplets. Furthermore, in R-Auto (Figure 2c), the AC electric field is applied to the process of washing out the antibody solutions, which was previously performed manually (reprinted with permission from Ref. [7]. Copyright 2017 K. Terata et al. Licensed under Creative Commons Attribution 4.0 International License (http://creativecommons.org/licenses/by/4.0/ (15 January 2025))).

The details of the staining method are listed in Table 1. For deparaffinization and antigen activation, the samples were first immersed three times for 3 min each in a solution of 85% xylene and 10% ethylbenzene. This was followed by three immersions for 3 min each in 99.5% ethanol. Thereafter, the samples were immersed in distilled water for 5 min and then treated with a heated antigen activation solution (ULTRA Cell Conditioning Solution (ULTRA CC2) Roche Diagnostics Inc., Tokyo, Japan) at 98 °C for 40 min. Finally, the samples were left at room temperature for 20 min to complete the antigen activation process. The pretreatment was completed in 83 min.

For the IHC antigen–antibody reaction, the slides were first incubated for 6 min with the anti-ER antibody (CONFIRM anti-estrogen receptor (ER) (SP1) rabbit monoclonal primary antibody, Roche Diagnostics Inc., Tokyo, Japan) in R-Auto. The primary antibody was then detected by incubating the slides for 4 min with a secondary antibody (ultraView Universal DAB Detection Kit, Roche Diagnostics Inc., Tokyo, Japan). As in previous reports [4,5,6,7,8,9,11], the application of an AC electric field reduced the reaction time needed for the primary antibody from 32 min to 6 min compared with conventional IHC and the commercial autostainer. Similarly, the reaction time of the secondary antibody was reduced from 32 min to 4 min compared with conventional IHC and from 12 min to 4 min compared with the commercial autostainer. An additional unique feature of R-Auto is that an AC electric field is applied for the washing process, which reduces the time required for washing out the antibody from 10 min (as needed for the conventional method) to 5 min. As a result, the total time required for IHC was significantly reduced from 209 min with conventional IHC and 210 min with the commercial autostainer to 121 min with the R-Auto method.

Finally, the slides were developed using 3,3′-diaminobenzidine (Dako liquid DAB+ Substrate Chromogen System, Dako, Tokyo, Japan) and counterstained with hematoxylin. The slides were incubated in Bluing Reagent (Roche Diagnostics Inc., Tokyo, Japan) and Ventana universal DAB copper (included in the ultraView Universal DAB Detection Kit) before and after counterstaining. The R-Auto protocol was established by setting the reaction times of the primary and secondary antibodies in 1 min increments and performing test staining. The protocol was subsequently judged by a pathologist to produce the best diagnostic performance. With the previously reported equipment for R-IHC, only the primary and secondary antibody reactions were performed in the automated equipment. However, with R-Auto, all steps in the protocol, including washing out of the antibody solution, were automated.

### 2.4. Histopathological Evaluation

Assigning Allred scores, which combines five levels of positive cell occupancy (proportion score (PS)) and three levels of staining intensity (intensity score (IS)), is a typical scoring method [12] for assessing the ER status of breast cancers. The proportion score has six levels: 0 (all negative), 1 (stained area < 1/100), 2 (1/100 to 1/10), 3 (1/10 to 1/3), 4 (1/3 to 2/3), and 5 (≥2/3). The intensity score has four levels: 0 (negative), 1 (weak), 2 (intermediate), and 3 (strong). The total score is classified into eight levels, ranging from 0 to 8. A total Allred score (TS) of 0-2 was classified as ER-negative, 3–6 as weakly positive, and 7–8 as strongly positive [13]. Pathological diagnoses were made by two certified pathologists.

### 2.5. Statistical Analysis

The diagnostic concordances between R-Auto and conventional IHC and between R-Auto and the commercial autostainer were evaluated. The number of required cases was calculated to be 214 (192 cases excluding specimen deficiencies) for 80% power when tested at a significance level of 0.05, assuming an expected concordance rate of 0.8 and a dropout rate of 10% due to incomplete specimens. Because evaluation bias was expected, the AC_1_ statistic [14], in addition to the weighted kappa coefficient, was used to evaluate the degree of agreement. The AC_1_ statistic is a measure used to assess the level of agreement between evaluators. It was developed as an alternative to the kappa coefficient [15], which can be difficult to use in certain situations. While the kappa coefficient adjusts for the probability of chance agreement, it tends to yield lower values when the agreement rate is high, which can be problematic. The AC_1_ statistic addresses this issue by accounting for the probability of chance agreement in its calculation. As a result, it provides more stable reliability, even in cases of high agreement rates. The cutoffs for the weighted kappa coefficient and the AC_1_ statistic were defined as poor for less than 0.40, moderate for 0.40 to 0.60, and substantive agreement for 0.60 or more. Statistical analysis was performed using the SAS 9.4 software (SAS Institute, Cary, NC, USA).

## 3. Results

The characteristics of the 188 patients evaluated are shown in Table 2. Based on the pathological diagnostics at surgery, most (70.2%) were diagnosed with invasive carcinoma NST. The staging revealed that 53.2% were stage I, 21.3% were stage II, and 10.2% were stage III. The ER immunostaining results (synonymous with the commercial autostainer staining results) at the time of postoperative diagnosis were TS0–2 in 28 specimens (14.9%), TS3–6 in 11 specimens (5.9%), and TS7–8 in 149 specimens (79.2%).

The staining profile obtained with R-Auto was TS0–2 in 28, TS3–6 in 13, and TS7–8 in 147 specimens; with conventional IHC, the corresponding staining profile was 28, 5, and 155 specimens; and for postoperative diagnosis with the commercial autostainer, it was 28, 11, and 149 specimens, respectively (Table 3 and Table 4). The AC_1_ statistic for comparison between R-Auto and conventional IHC was 0.9490 (0.9139–0.9841), with a 95.7% agreement rate. For comparison between R-Auto and the commercial autostainer, the AC_1_ statistic was 0.9095 (0.8620–0.9570), with a 92.6% agreement rate. Thus, the AC_1_ statistics showed substantial agreement between the results obtained using R-Auto and both the conventional staining method and staining using the commercial autostainer. Moreover, when we separately compared the agreement rate for each score from TS0 to TS8 instead of the three-level scoring, the AC_1_ statistic was 0.9160 (0.8749–0.9572) for R-Auto vs. conventional IHC and 0.8372 (0.7816–0.8927) for R-Auto vs. the commercial autostainer. There was substantial agreement. The microscopic images show that the samples were stained very well and clearly with R-Auto and nearly equally (Figure 3).

## 4. Discussion

In this study, there was strong agreement between the Allred scores obtained using R-Auto, conventional IHC, and the commercial autostainer. In addition, R-Auto reduced the antigen–antibody reaction time from 64 and 44 min to 10 min compared with conventional IHC and commercial autostainer, respectively. Additionally, R-Auto reduced the washing time from 28 min to 15 min compared with conventional IHC. The time reduction and good staining performance are attributable to the activation of the antigen–antibody reaction using an AC electric field, which also sped up the washing process.

Although good agreement was obtained, some differences in the staining results obtained with the commercial autostainer and R-Auto were noted. In that regard, a study examining the staining of old FFPE specimens reported no differences in the IHC staining intensity of cytoplasmic antigens in old specimens, but the staining of the membrane and nuclear antigens declined over time [16]. It has also been reported that the intensity of ER staining differs depending on the storage method [17]. In this study, the staining intensity may have been affected by the use of thin slices of the specimens that were prepared several years after the tissue blocks were made, and this may have limited obtaining the most accurate results possible.

IHC has several limitations. In basic research, IHC involves the use of a wide variety of antibodies. Since it is difficult to explore the optimal conditions for all of them, the reproducibility of the experimental results decreases. Variability in results occurs between laboratories and researchers, and this is considered a problem as it leads to prolonged research periods and significant costs [18]. Additionally, in pathological diagnosis, it is also a problem that the standardization of IHC protocols and the assurance of reproducibility are not ensured between institutions [19,20]. R-IHC makes the staining intensity reproducible by applying an AC electric field, and R-Auto can further standardize protocols. These features can contribute to solving such issues in both research laboratories and hospitals.

Currently, the use of IHC in determining treatment strategies for breast cancer is influenced by gene expression profiles (GEPs). Perou et al. performed GEPs of breast cancer using cDNA microarrays and proposed an intrinsic subtype classification based on GEPs in 2000 [18,19,20]. In this classification, breast cancer was divided into subtypes with different biological characteristics, such as luminal A, luminal B, HER2-enriched, basal-like, normal breast-like, etc. The subtypes have different prognoses and drug resistance. Although GEPs are useful in that they can serve as an indicator of drug therapy selection, it is not clinically realistic to perform GEPs in all breast cancer patients. Therefore, since 2011, an alternative definition of intrinsic subtype based on pathological ER/PgR/HER2/Ki67 status has been proposed [21,22,23,24,25]. However, treatment strategies cannot always be determined by the IHC alone. In ER-positive HER2-negative breast cancer especially, it is recommended to use multi-gene assays, such as Oncotype DX [26], which assess the additional effects of chemotherapy and prognostic predictions based on the intrinsic subtype, to decide on the treatment strategy. The results from such assays take approximately one month to be returned. In addition to ER, HER2 is also determined as positive or negative using ISH, which requires approximately two weeks when outsourced. Therefore, it is crucial to quickly complete the initial IHC. A study of 5137 breast cancer patients requiring neoadjuvant chemotherapy reported that the estimated 5-year overall survival significantly worsened as the number of days from diagnosis to intervention increased [27]. In recent years, the development of novel systemic therapies, such as antibody–drug conjugates [28] and immune checkpoint inhibitors [29,30,31,32,33,34], has progressed, leading to an increasing number of IHC tests required for selecting cancer drug therapies [35]. There is a growing demand for rapid and highly accurate IHC.

The limitation of this study is that we only evaluated ER among the IHCs required for breast cancer. Previous studies have shown that PgR expression fluctuates according to the menstrual cycle [36]. In the data used in this study, there was no information available about the patients’ menstrual statuses at the time of surgery, which may have hindered an accurate assessment of PgR expression. Additionally, if HER2 scores are 2+, further testing using IHC or ISH methods is required for accurate evaluation, which delays treatment initiation as we must wait for the results. For this reason, we decided to focus on ER first. Meanwhile, a key feature of R-Auto is its universal applicability. R-IHC protocols for HER2 and Ki67 on frozen sections have already been reported [5,8]. These protocols do not significantly differ in antigen–antibody reaction times from the ER protocols evaluated in this study. By applying these findings, we anticipate that we will be able to smoothly evaluate other IHC tests for R-Auto on FFPE sections.

In this study, we showed that R-Auto can perform IHC with sufficient staining accuracy in the same or less time than commercial autostainers. We suggest R-Auto could be used in both clinical practice and the laboratory to provide high-quality, reproducible IHC results. In addition, R-Auto reduces the amount of the expensive antibody needed by activating the antigen–antibody reaction with AC electric field agitation [4,8]. Furthermore, we anticipate that it will also be applicable to in situ hybridization [11]. Although we stained only for ER in this study, a strength of this method is that it could also be applied to PgR, HER2, and Ki-67, which are important biomarkers in breast cancer treatment. Additionally, it could be used to expedite immunostaining in other types of cancer. Furthermore, with the potential spread of digital pathology in daily practice, it is essential to work toward standardizing and streamlining IHC. We believe that our technology can contribute to this effort.

## 5. Conclusions

We developed a fully automated R-IHC staining machine that achieves rapid, high-quality IHC staining with little human effort via the application of an AC electric field to enhance the antibody–antigen interaction and the washing process. This enables the establishment of an ER immunostaining protocol in breast cancer that could be the basis for the use of R-IHC in many other cancers.

## Figures and Tables

**Figure 1 cancers-17-00363-f001:**
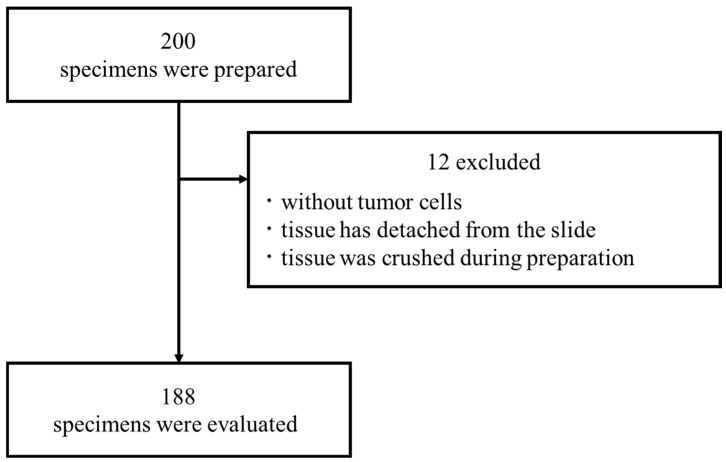
Data collection and exclusion criteria.

**Figure 2 cancers-17-00363-f002:**
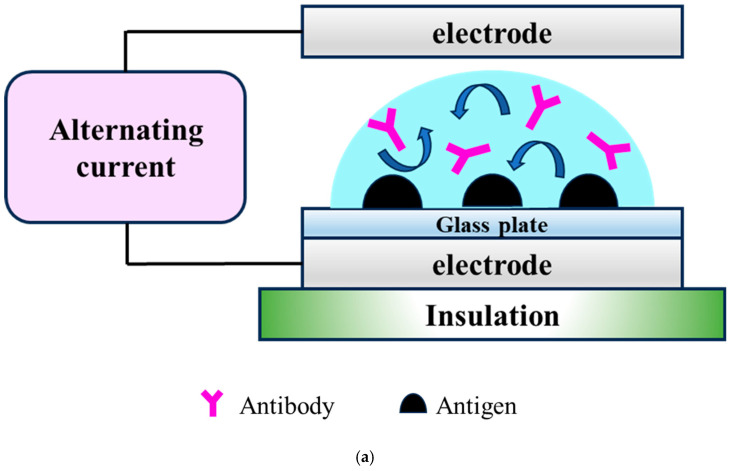
Principles of rapid immunohistochemistry. (**a**) The slide was placed between the electrodes, and a high-voltage (4 KV), low-frequency (5 Hz) alternating current (AC) was applied. Reprinted with permission from Ref. [7]. Copyright 2017 K. Terata et al. Licensed under Creative Commons Attribution 4.0 International License (http://creativecommons.org/licenses/by/4.0/ (15 January 2025)). The illustration (a) was modified based on Ref. [7]. (**b**) The changes within a microdroplet as the voltage was switched on and off mixed the antibodies. Reprinted with permission from Ref. [7]. Copyright 2017 K. Terata et al. Licensed under Creative Commons Attribution 4.0 International License (http://creativecommons.org/licenses/by/4.0/ (15 January 2025)). (**c**) The fully automated rapid immunohistochemistry stainer (R-Auto).

**Figure 3 cancers-17-00363-f003:**
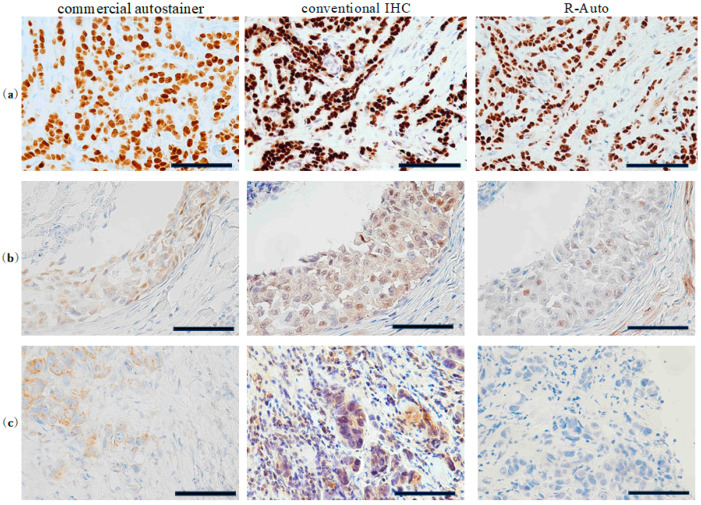
Image showing immunoreactivity of ER expressed in the nucleus of breast cancer. Tissue sections were stained with hematoxylin and anti-ER antibody. (**a**) All matching cases of TS7 or -8. (**b**) All matching cases of TS3–6. (**c**) All matching cases of TS0–2. Scale bar represents 250 μm. ER, estrogen receptor. TS, total score. Magnification of objective lens is 40×.

**Table 1 cancers-17-00363-t001:** The procedures of the commercial autostainer, conventional IHC, and R-Auto.

	Time (Minutes)		
Procedure	Commercial Autostainer	Conventional IHC	R-Auto
Pretreatment *	64	83	
Blocking endogenous peroxidase activity		10	3
Washing with PBS	- *	6	
Primary antibody	32	32	6 **
Washing with PBS	- *	10	5 **
Ventana ultraView Universal DAB Detection Kit	12	32	4 **
Washing with PBS	- *	10	5 **
3,3′-diaminobenzidine	- *	5	6
Washing with PBS	- *	2	2
Hematoxylin nuclear counter staining	12	1	11
Approximate time required ****	- *	108 ***	42 ***
Dehydrating, permeating, and sealing	- *	18	
Approximate overall time required *****	210	209	121

*: not disclosed; **: with AC electric field (4 _K_V; 10 Hz); ***: approximate time required for IHC excluding pretreatment; PBS: phosphate-buffered saline; ****: time required for dyeing; *****: total time required including all processes; DAB: 3,3′-diaminobenzidine. The concentration of the solution used for pretreatment: a solution of 85% xylene and 10% ethylbenzene.

**Table 2 cancers-17-00363-t002:** Clinical details of these breast cancer patients.

		n = 188
Age	Median (Range)	62 (21–91)
Histological type	Invasive carcinoma NST	132 (70.2%)
	Lobular	10 (5.3%)
	Intraductal	28 (14.9%)
	Others	18 (9.6%)
Histological grade	I	61 (32.4%)
	II	93 (49.5%)
	III	24 (12.8%)
	Unknown	10 (5.3%)
Vessel invasion	No	150 (79.8%)
	Yes	36 (19.1%)
	Unknown	2 (1.1%)
Lymphatic invasion	No	100 (53.2%)
	Yes	86 (42.6%)
	Unknown	2 (1.1%)
Stage	0	28 (14.9%)
	I	100 (53.2%)
	IIA	30 (16.0%)
	IIB	10 (5.3%)
	IIIA	9 (4.8%)
	IIIB	8 (4.3%)
	IIIC	2 (1.1%)
	Unknown	1 (0.5%)
ER (Allred score)	Total score 7–8	149 (79.2%)
	3–6	11 (5.9%)
	0–2	28 (14.9%)
PgR (Allred score)	Total score 7–8	149 (79.2%)
	3–6	11 (5.9%)
	0–2	28 (14.9%)
HER2 score	0	55 (29.3%)
	1+	44 (23.4%)
	2+	67 (35.6%)
	3+	22 (11.7%)
Ki67	20<	124 (66.0%)
	≦20	64 (34.0%)

**Table 3 cancers-17-00363-t003:** Comparison of diagnostic results between R-Auto and conventional IHC.

		Conventional IHC			
	Allred Score	TS0–2	3–6	7–8	Total
R-Auto	TS0–2	28	0	0	28
	3–6	0	5	8	13
	7–8	0	0	147	147
	Total	28	5	155	188
		Estimated Value		95% Confidence Interval	
Kappa coefficient		0.8716		0.7859–0.9572	
Weighted kappa coefficient		0.9254		0.8727–0.9782	
AC_1_ statistics		0.9490		0.9139–0.9841	
Agreement		95.7%			

**Table 4 cancers-17-00363-t004:** Comparison of diagnostic results between R-Auto and commercial autostainer.

		Commercial Autostainer			
	Allred Score	TS0–2	3–6	7–8	Total
R-Auto	TS0–2	23	3	2	28
	3–6	5	6	2	13
	7–8	0	2	145	147
	Total	28	11	149	188
		Estimated Value		95% Confidence Interval	
Kappa coefficient		0.7897		0.6938–0.8856	
Weighted kappa coefficient		0.8554		0.7814–0.9294	
AC_1_ statistics		0.9095		0.8620–0.9570	
Agreement		92.6%			

## Data Availability

Samples and information related to the research are managed with a unique identification number and anonymized, with full consideration given to protecting the confidentiality of the research participants. The data are stored offline in a database, kept in a locked location within our department, and shared with all authors.

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
