# Peer review of "Evaluation of Grading Estrogen Receptors in Breast Cancer Using Fully Automated Rapid Immunohistochemistry Based on Alternating-Current Electric Field Technology"

_cancers, 2025, doi:10.3390/cancers17030363_

Round 1
Reviewer 1 Report
Comments and Suggestions for Authors
The quality of the article is above average. The presented results are clear and have statistical value.
I consider that there are some small changes to be made to this article. Some sentences are very long and lead to the loss of the main idea " IHC analyses of estrogen receptor (ER), progesterone receptor
(PgR), human epidermal growth factor receptor type 2 (HER2) and the Ki-67 index are
used as surrogates for genetic profiling to determine breast cancer treatment strategies,
which include neoadjuvant or adjuvant systemic therapy to reduce the risk of recurrence,
to prolong life and to maintain quality of life."
I recommend that the authors move the paragraph from lines 192-204 to the Method chapter. The content of this paragraph refers to the method used and not to the results obtained.
There is a calculation error in Table no. 2 /line 4 -Lobular 10 (53.2%)
Author Response
The authors thank editor and reviewers for carefully reading our revised manuscript and for these thoughtful comments. The reviewer’s comments are in bold, and our responses follow each comment.
The quality of the article is above average. The presented results are clear and have statistical value.
I consider that there are some small changes to be made to this article. Some sentences are very long and lead to the loss of the main idea " IHC analyses of estrogen receptor (ER), progesterone receptor (PgR), human epidermal growth factor receptor type 2 (HER2) and the Ki-67 index are used as surrogates for genetic profiling to determine breast cancer treatment strategies,which include neoadjuvant or adjuvant systemic therapy to reduce the risk of recurrence,to prolong life and to maintain quality of life."
I recommend that the authors move the paragraph from lines 192-204 to the Method chapter. The content of this paragraph refers to the method used and not to the results obtained.
There is a calculation error in Table no. 2 /line 4 -Lobular 10 (53.2%)
Thank you very much for your comment.
I agree with your suggestion that this section might be better moved to the Method chapter. I have made some adjustments accordingly sentence to line 183-193.
Additionally, the correct value for the calculation was 5.3%, so I have made the necessary correction in Table 2 . It is the 4th row from the top in Table 2, where the highlights are applied.
I have adopted the main idea you recommended and added it to the manuscript in the following location. I have added this sentence to line 90-94.
Reviewer 2 Report
Comments and Suggestions for Authors
The aim of this manuscript was to evaluate the clinical reliability of the R-Auto protocol for staining estrogen receptors, progesterone receptor, human epidermal growth factor receptor type 2 and the Ki-67 in breast cancer specimens and evaluated the staining performance, in 188 surgical specimens, using R-Auto, conventional manual (conventional) immunohistochemistry and a commercial autostainer. They have developed a fully automated immunohistochemistry staining machine (R-IHC) that achieves rapid (shorter time with less effort), high quality immunohistochemistry staining with little human effort through application of an alternating current electric field to enhance the antibody-antigen interaction and the washing process, focusing on establishment of an ER immunostaining protocol in breast cancer that could serve as the basis for the use of R-IHC in many other cancers. I have some minor comments on this:
1.- Since the work is about to establish a staining protocol and evaluate the staining performance of R-Auto against in proteins for FFPE samples of breast cancer. There is no point in writing down a paragraph of the treatment strategies or drugs, they could rather discuss about other technologies such as tissue arrays, or why the need to shorten the times of performing immunohistochemistry. They don't even take it into account in the discussion
2.- The resolution of figure 2-C needs to be improved.
3.- The footnote in figure 2 a to c, although it contains intuitive and/or known elements, these do not describe what each figure, arrow, shape or color represents. It is necessary to make a more detailed description.
4.- In the methods section, it is important to note the brands, catalog and degree of purity used in the IHC, for example xylene, ethanol, etc. In the same section, it is not necessary to note the registered trademark symbol ®, since they are not the owners of it (table 1).
5.- Regarding figure 3, you must note the magnification of the microscope objective, and in the text describe how you calculate the intensity, whether by crosses or with a program.
6.- Provide a little detail on the tests used in statistical analysis
7.- Check spelling in general, for example minites (line 201). They should also note acronyms, for example DAB or AC, even if they are well-known or universal.
8.-Perhaps they could add as supplementary figures the IHC of progesterone receptor, human epidermal growth factor receptor type 2 or Ki-67. In general, they focus on the discussion of the estrogen receptor, if they are not going to discuss or compare the other proteins there is no point in them appearing in some sections of the manuscript.
9.- It is necessary that they review the annotations throughout the manuscript, for example, in table 2 (line 211) histological type, lobular, they note 10 patients equal to 53.2%, or in Ki67 why 20< and ≦20, how is it different?
10.- In table 4, what is the meaning of the symbol 計 (total?)
11.- To describe whether immunohistochemistry times with R-IDC, between estrogen receptor (ER), progesterone receptor (PgR), human epidermal growth factor receptor type 2 (HER2) and Ki-67 vary or are equal?
Author Response
The authors thank editor and reviewers for carefully reading our revised manuscript and for these thoughtful comments. The reviewer’s comments are in bold, and our responses follow each comment.
‐1.- Since the work is about to establish a staining protocol and evaluate the staining performance of R-Auto against in proteins for FFPE samples of breast cancer. There is no point in writing down a paragraph of the treatment strategies or drugs, they could rather discuss about other technologies such as tissue arrays, or why the need to shorten the times of performing immunohistochemistry. They don't even take it into account in the discussion.
Thank you very much for your comments. As you suggested, the discussion about other technologies and the need to shorten the time required for IHC was very important, so I added it to the Introduction and Discussion part.
The following was added to the line 57-59 as the time required for conventional immunostaining and its limitation.
“However, standard IHC procedures in frozen or paraffin sections take 2–4 hr, so they cannot be used for intraoperative frozen section diagnosis”
The following was added to the line 69-74 as comparison with other techniques.
“Similarly, there have been reports on developments related to rapid IHC. For example, the literature reported that the rapid IHC method using the intermittent microwave irradiation method (MW) makes it possible to reduce the entire immunohistochemical staining process to one hour [9]. However, because this method uses microwaves, there is concern about tissue and cell degeneration due to high-temperature heat. Our technology has an advantage in this respect.”
The following was added to the line 300-309 as the limitation of conventional IHC technique.
“IHC has several limitations. In basic research, IHC involves the use of a wide variety of antibodies. Since it is difficult to explore the optimal conditions for all of them, the reproducibility of the experimental results decreases. Variability in results occurs between laboratories and researchers, and this is considered a problem as it leads to prolonged research periods and significant costs [18]. Additionally, in pathological diagnosis, it is also a problem that the standardization of IHC protocols and the assurance of reproducibility are not ensured between institutions [19.20]. R-IHC makes the staining intensity reproducible by applying an AC electric field, and R-Auto can fur-ther standardize protocols. These features can contribute to solving such issues in both research laboratories and hospitals.”
The following was added to the line 310-332 as other techniques such as multigene assay and the reason why we need rapid IHC technology.
“Currently, the use of IHC in determining treatment strategies for breast cancer is influenced by gene expression profiles (GEPs). Perou et al. performed GEPs of breast cancer using cDNA microarrays and proposed an intrinsic subtype classification based on GEPs in 2000 [18-20]. In this classification, breast cancer was divided into subtypes with different biological characteristics, such as luminal A, luminal B, HER2-enriched, basal-like, normal breast-like, etc. The subtypes have different prognoses and drug re-sistance. Although GEPs are useful in that they can serve as an indicator of drug ther-apy selection, it is not clinically realistic to perform GEPs in all breast cancer patients. Therefore, since 2011, an alternative definition of intrinsic subtype based on patholog-ical ER/PgR/HER2/Ki67 status has been proposed [21-25]. However, treatment strate-gies cannot always be determined by the intrinsic subtype alone. Especially in ER-positive HER2-negative breast cancer, it is recommended to use multi-gene assays, such as Oncotype DX [26], which assess the additional effects of chemotherapy and prognostic predictions based on the intrinsic subtype, to decide on the treatment strat-egy. The results from such assays take approximately one month to be returned. In ad-dition to ER, HER2 is also determined as positive or negative using ISH, which requires approximately two weeks when outsourced. Therefore, it is crucial to quickly complete the initial IHC. A study of 5137 breast cancer patients requiring neoadjuvant chemo-therapy reported that the estimated 5-year overall survival significantly worsened as the number of days from diagnosis to intervention increased [27]. In recent years, the development of novel systemic therapies, such as antibody–drug conjugates [28] and immune checkpoint inhibitors [29-34], has progressed, leading to an increasing num-ber of IHC tests required for selecting cancer drug therapies [35]. There is a growing demand for rapid and highly accurate IHC.”
-2.- The resolution of figure 2-C needs to be improved.
Thank you very much for your comments.I have increased the resolution about line 157. If further improvements are needed, I will address them promptly.
-3.- The footnote in figure 2 a to c, although it contains intuitive and/or known elements, these do not describe what each figure, arrow, shape or color represents. It is necessary to make a more detailed description.
Thank you very much for your comment.
This is a schematic diagram showing the pathological tissue on the slide glass and the agitation of antibody droplets in the electric field. In response to your suggestion, I have added the figure and line 130-147.
“We have developed a device that reduces the time required for IHC analysis. Its mechanism has been previously described in detail [6,11]. Briefly, in this R-IHC system, a high-voltage, low-frequency (4KV; 10Hz) alternating current (AC) electric field is applied to tissue sections while they are incubating with the antibodies. The schematic diagram of the device used to apply the high-voltage, low-frequency AC electric field is shown (Fig. 2a, b). The slide is placed between the electrodes, and a high-voltage (4 KV), low-frequency (5 Hz) AC current is applied. The arrows in each figure indicate that the antibodies are being stirred due to the vibration of the droplets. “Schema of the device used to apply a high-voltage, low-frequency AC electric field. The slide was placed between the electrodes, and a high-voltage (4 KV), low-frequency (5 Hz) AC current was applied . The schema shows the changes within a microdroplet as the voltage is switched on and off in a time series (I → II → III → IV), which mixes the antibodies. “(This sentence is cited from reference [7].)(Fig. 2b). This significantly shortens the time required for the antigen–antibody reaction. The AC electric field is effective for stirring liquids in the order of µL and is an ideal technique for IHC as it does not increase the temperature of the droplets. Furthermore, in R-Auto (Fig. 2c), the AC electric field is applied to the process of washing out the antibody solutions, which was previously performed manually. Figure 2b is cited from reference [7].”
And I have added the same content to the caption of Figure 2 as well , line 164.
-4.- In the methods section, it is important to note the brands, catalog and degree of purity used in the IHC, for example xylene, ethanol, etc. In the same section, it is not necessary to note the registered trademark symbol ®, since they are not the owners of it (table 1).
Thank you very much for your comment. The issue had already been corrected by the editor during the preprint stage.
I appreciate your assistance in pointing it out.
-5.- Regarding figure 3, you must note the magnification of the microscope objective, and in the text describe how you calculate the intensity, whether by crosses or with a program.
Thank you very much for your comment.
In response, I have added the magnification of the objective lens (40x) to the manuscript at line 281 .
Additionally, I have described the calculation method for the Allred score.
The evaluation is based on the sum of two scores: the Proportion Score, which represents the percentage of stained cells within the total cell population, and the Intensity Score, which indicates the staining intensity.
“The Proportion Score has six levels: 0 (all negative), 1 (stained area < 1/100), 2 (1/100 to 1/10), 3 (1/10 to 1/3), 4 (1/3 to 2/3), and 5 (≥2/3). The Intensity Score has four levels: 0 (negative), 1 (weak), 2 (intermediate), and 3 (strong). The total score is classified into eight levels, ranging from 0 to 8.”
This explanation has been added to the manuscript at line 208-211.
2-6.- Provide a little detail on the tests used in statistical analysis.
Thank you very much for your comment.
The AC1 statistic is a measure of inter-rater agreement, developed as an alternative to Cohen’s kappa coefficient (κ) when the latter is difficult to use.
The κ coefficient adjusts for the probability of agreement occurring by chance, but it tends to yield low values when agreement rates are high, which is a known limitation. The AC1 statistic addresses this issue by accounting for the probability of chance agreement, providing a more stable reliability measure even in scenarios of high agreement.
Since the purpose of this study was to confirm a high degree of agreement, we also used the AC1 statistic.
“Because evaluation bias was expected, the AC1 statistic [14], in addition to the weighted kappa coefficient, was used to evaluate the degree of agreement. The AC1 statistic is a measure used to assess the level of agreement between evaluators. It was developed as an alternative to the kappa coefficient [15], which can be difficult to use in certain situations. While the kappa coefficient adjusts for the probability of chance agreement, it tends to yield lower values when the agreement rate is high, which can be problematic. The AC1 statistic addresses this issue by accounting for the probability of chance agreement in its calculation. As a result, it provides more stable reliability even in cases of high agreement rates.The cutoffs for the weighted kappa coefficient and the AC1 statistic were defined as poor for less than 0.40, moderate for 0.40 to 0.60, and substantive agreement for 0.60 or more. Statistical analysis was performed using the SAS 9.4 software (SAS Institute, Cary, NC, USA).”
This explanation has been added to the manuscript at line 220-231.
-7.- Check spelling in general, for example minites (line 201). They should also note acronyms, for example DAB or AC, even if they are well-known or universal.
Thank you very much for your comment.
I have made the necessary revisions and added explanations for terms such as AC and DAB in the footnote of Table 1, line 175-178.
-8.-Perhaps they could add as supplementary figures the IHC of progesterone receptor, human epidermal growth factor receptor type 2 or Ki-67. In general, they focus on the discussion of the estrogen receptor, if they are not going to discuss or compare the other proteins there is no point in them appearing in some sections of the manuscript.
Thank you very much for your comment.
In this study, we focused on ER, and no staining for other markers such as PgR was performed.
Therefore, I have removed any related descriptions from the manuscript.
-9.- It is necessary that they review the annotations throughout the manuscript, for example, in table 2 (line 211) histological type, lobular, they note 10 patients equal to 53.2%, or in Ki67 why 20< and ≦20, how is it different?
Thank you for your comment.The 53.2% was an error, and I will correct it, about line 575.
The evaluation method for Ki67 staining has some ambiguity, as some institutions assess hotspots, while others evaluate the overall average, leading to variability in results.
In a retrospective study of the Australian Breast and Colorectal Cancer Study Group (ABCSG) Trial 5, which focused on premenopausal hormone receptor-positive early-stage breast cancer, using 20% as the cutoff for Ki67 showed that the high Ki67 group had significantly shorter relapse-free survival and overall survival compared to the low Ki67 group1).
Additionally, there have been reports indicating that the risk of distant metastasis is significantly higher when Ki67 is over 20%. Some reports suggest that a cutoff of 14% may also be appropriate2).
Many clinical trials for breast cancer have found a correlation between a Ki67 cutoff line of 20%-25% and prognosis, and it is customary to use 20% as the cutoff.
Therefore, we classified the samples accordingly in this study.
I will attach the references for your review.
- Bago-Horvath Z, Rudas M, Singer CF, Greil R, Balic M, Lax SF, et al. Predictive value of molecular subtypes in premenopausal women with hormone receptor-positive early breast cancer:results from the ABCSG trial 5. Clin Cancer Res. 2020;26(21):5682-8. [PMID:32546648]
- Focke CM, Bürger H, van Diest PJ, Finsterbusch K, Gläser D, Korsching E, et al;German Breast Screening Pathology Initiative. Interlaboratory variability of Ki67 staining in breast cancer. Eur J Cancer. 2017;84:219-27. [PMID:28829990]
-10.- In table 4, what is the meaning of the symbol 計 (total?)
Thank you very much for your comment.
The term "計" was intended to mean "total" in Japanese.
I appreciate your assistance in pointing it out.
11.- To describe whether immunohistochemistry times with R-IDC, between estrogen receptor (ER), progesterone receptor (PgR), human epidermal growth factor receptor type 2 (HER2) and Ki-67 vary or are equal?
Thank you very much for your comment.
Based on your feedback, I have added this information to 340-344 line.
As described below, there is very little difference.
“a key feature of R-Auto is its universal applicability. R-IHC protocols for HER2 and Ki67 on frozen sections have already been reported [5.8]. These protocols do not significantly differ in antigen–antibody reaction times from the ER protocols evaluated in this study.”
Reviewer 3 Report
Comments and Suggestions for Authors
This manuscript studies a new type of fully automated rapid immunohistochemistry (IHC) stainer (R-Auto), and evaluates its performance and clinical reliability in estrogen receptor (ER) staining of breast cancer specimens by comparing the traditional manual IHC method with a commercial automated stainer. The article is complete, scientifically rigorous, with clear research methods, detailed results and data, and in-depth discussions, and has certain clinical translation value. However, there are still some areas that need to be improved to improve the scientificity and logic of the article.
1. Although the background section of the manuscript mentions the importance of IHC in cancer diagnosis, it does not fully explain the specific limitations of existing IHC technology (such as time, cost, human error, etc.) and its impact on clinical practice. It is recommended to add specific data or examples to more clearly highlight the significance and innovation of R-Auto technology.
2. The technical details of the R-Auto staining process (such as the parameters of the AC electric field, the amount of antibodies, etc.) are relatively general, and it is recommended to further explain in detail so that readers can have a more comprehensive understanding of the technical principles and operation methods. The author should consider explaining it, otherwise it should at least be explained in the manuscript.
3. The manuscript mainly focuses on the advantages of R-Auto, but the universality of its applicability to other antibody markers is insufficient (such as PgR, HER2, Ki-67). What do the authors think of this issue?
In short, the manuscript is very good and can be accepted after minnor revision.
Author Response
The authors thank editor and reviewers for carefully reading our revised manuscript and for these thoughtful comments. The reviewer’s comments are in bold, and our responses follow each comment.
This manuscript studies a new type of fully automated rapid immunohistochemistry (IHC) stainer (R-Auto), and evaluates its performance and clinical reliability in estrogen receptor (ER) staining of breast cancer specimens by comparing the traditional manual IHC method with a commercial automated stainer. The article is complete, scientifically rigorous, with clear research methods, detailed results and data, and in-depth discussions, and has certain clinical translation value. However, there are still some areas that need to be improved to improve the scientificity and logic of the article.
3-1 Although the background section of the manuscript mentions the importance of IHC in cancer diagnosis, it does not fully explain the specific limitations of existing IHC technology (such as time, cost, human error, etc.) and its impact on clinical practice. It is recommended to add specific data or examples to more clearly highlight the significance and innovation of R-Auto technology.
Thank you very much for your comment.
As you suggested, the discussion about other technologies and the need to shorten the time required for IHC was very important, so I added it to the Introduction and Discussion part.
The following was added to lines 57-59 to address the issues with conventional IHC techniques.
“The immunohistochemistry (IHC) technique uses antibodies to detect specific an-tigens within samples. Combining antigen–antibody reactions with a color reaction enables the presence and localization of an antigen within a tissue sample to be ob-served under a microscope, thereby confirming the expression of specific genes or pro-teins. For cancer treatment, IHC helps make differential diagnoses of benign and ma-lignant tumors, visualize the extent of cancers, and select drug therapies. However, standard IHC procedures in frozen or paraffin sections take 2–4 hr, so they cannot be used for intraoperative frozen section diagnosis [1,2,3].”
The following was added to the line 300-309 as the limitation of conventional IHC technique.
“IHC has several limitations. In basic research, IHC involves the use of a wide variety of antibodies. Since it is difficult to explore the optimal conditions for all of them, the reproducibility of the experimental results decreases. Variability in results occurs between laboratories and researchers, and this is considered a problem as it leads to prolonged research periods and significant costs [18]. Additionally, in pathological diagnosis, it is also a problem that the standardization of IHC protocols and the assurance of reproducibility are not ensured between institutions [19.20]. R-IHC makes the staining intensity reproducible by applying an AC electric field, and R-Auto can further standardize protocols. These features can contribute to solving such issues in both research laboratories and hospitals.”
The following was added to the line 310-332 as other techniques such as multigene assay and the reason why we need rapid IHC technology.
“Currently, the use of IHC in determining treatment strategies for breast cancer is influenced by gene expression profiles (GEPs). Perou et al. performed GEPs of breast cancer using cDNA microarrays and proposed an intrinsic subtype classification based on GEPs in 2000 [18-20]. In this classification, breast cancer was divided into subtypes with different biological characteristics, such as luminal A, luminal B, HER2-enriched, basal-like, normal breast-like, etc. The subtypes have different prognoses and drug re-sistance. Although GEPs are useful in that they can serve as an indicator of drug ther-apy selection, it is not clinically realistic to perform GEPs in all breast cancer patients. Therefore, since 2011, an alternative definition of intrinsic subtype based on patholog-ical ER/PgR/HER2/Ki67 status has been proposed [21-25]. However, treatment strate-gies cannot always be determined by the intrinsic subtype alone. Especially in ER-positive HER2-negative breast cancer, it is recommended to use multi-gene assays, such as Oncotype DX [26], which assess the additional effects of chemotherapy and prognostic predictions based on the intrinsic subtype, to decide on the treatment strat-egy. The results from such assays take approximately one month to be returned. In ad-dition to ER, HER2 is also determined as positive or negative using ISH, which requires approximately two weeks when outsourced. Therefore, it is crucial to quickly complete the initial IHC. A study of 5137 breast cancer patients requiring neoadjuvant chemo-therapy reported that the estimated 5-year overall survival significantly worsened as the number of days from diagnosis to intervention increased [27]. In recent years, the development of novel systemic therapies, such as antibody–drug conjugates [28] and immune checkpoint inhibitors [29-34], has progressed, leading to an increasing num-ber of IHC tests required for selecting cancer drug therapies [35]. There is a growing demand for rapid and highly accurate IHC.”
The technical details of the R-Auto staining process (such as the parameters of the AC electric field, the amount of antibodies, etc.) are relatively general, and it is recommended to further explain in detail so that readers can have a more comprehensive understanding of the technical principles and operation methods. The author should consider explaining it, otherwise it should at least be explained in the manuscript.
Thank you very much for your comment .
This technique generates a suction force by applying a high-voltage electric field in a non-contact environment to droplets placed between parallel plates, which allows for the control of the droplet behavior and the movement of the substances inside the droplets, thus facilitating stirring.
By positioning the slide between the electrodes and applying an alternating current (AC) of high voltage (4 kV) and low frequency (5 Hz), the droplet can be stirred due to the changes within the droplet as the electric field switches on and off.
I have added this explanation along with the schematic diagram to 130-147 line.
“We have developed a device that reduces the time required for IHC analysis. Its mechanism has been previously described in detail [6,11]. Briefly, in this R-IHC system, a high-voltage, low-frequency (4KV; 10Hz) alternating current (AC) electric field is applied to tissue sections while they are incubating with the antibodies. The schematic diagram of the device used to apply the high-voltage, low-frequency AC electric field is shown (Fig. 2a, b). The slide is placed between the electrodes, and a high-voltage (4 KV), low-frequency (5 Hz) AC current is applied. The arrows in each figure indicate that the antibodies are being stirred due to the vibration of the droplets. “Schema of the device used to apply a high-voltage, low-frequency AC electric field. The slide was placed between the electrodes, and a high-voltage (4 KV), low-frequency (5 Hz) AC current was applied . The schema shows the changes within a microdroplet as the voltage is switched on and off in a time series (I → II → III → IV), which mixes the antibodies. “(This sentence is cited from reference [7].)(Fig. 2b). This significantly shortens the time required for the antigen–antibody reaction. The AC electric field is effective for stirring liquids in the order of µL and is an ideal technique for IHC as it does not increase the temperature of the droplets. Furthermore, in R-Auto (Fig. 2c), the AC electric field is applied to the process of washing out the antibody solutions, which was previously performed manually. Figure 2b is cited from reference [7].”
The manuscript mainly focuses on the advantages of R-Auto, but the universality of its applicability to other antibody markers is insufficient (such as PgR, HER2, Ki-67). What do the authors think of this issue?
Thank you very much for your comment .
We have identified the limitations of this study as follows and have added them to the manuscript, line 333-344.
“The limitation of this study is that we only evaluated ER among the IHCs required for breast cancer. Previous studies have shown that PgR expression fluctuates according to the menstrual cycle[36]. In the data used in this study, there was no information available about the patients' menstrual statuses at the time of surgery, which may have hindered an accurate assessment of PgR expression. Additionally, if HER2 scores are 2+, further testing using IHC or ISH methods is required for accurate evaluation, which delays treatment initiation as we must wait for the results. For this reason, we decided to focus on ER first. Meanwhile, a key feature of R-Auto is its universal applicability. R-IHC protocols for HER2 and Ki67 on frozen sections have already been reported [5.8]. These protocols do not significantly differ in antigen–antibody reaction times from the ER protocols evaluated in this study. By applying these findings, we anticipate that we will be able to smoothly evaluate other IHC tests for R-Auto on FFPE sections.”